# Auto Completion of User Interface Layout Design Using Transformer-Based Tree Decoders

## Abstract

It has been of increasing interest in the field to develop automatic machineries to facilitate the design process. In this paper, we focus on assisting graphical user interface (UI) layout design, a crucial task in app development. Given a partial layout, which a designer has entered, our model learns to complete the layout by predicting the remaining UI elements with a correct position and dimension as well as the hierarchical structures. Such automation will significantly ease the effort of UI designers and developers. While we focus on interface layout prediction, our model can be generally applicable for other layout prediction problems that involve tree structures and 2-dimensional placements. Particularly, we design two versions of Transformer-based tree decoders: Pointer and Recursive Transformer, and experiment with these models on a public dataset. We also propose several metrics for measuring the accuracy of tree prediction and ground these metrics in the domain of user experience. These contribute a new task and methods to deep learning research.

## 1 Introduction

Layout design is a universal task in many domains, ranging from mechanical design to graphical layouts. There has been a long tradition in developing computer-aided design (CAD) tools in both academia and industry [1](Hurst et al., 2009). Recently, there has been increasing interest in developing deep learning models for design generation (Zheng et al., 2019; Ha and Eck, 2017; Li et al., 2019; Isola et al., 2016). In this paper, we focus on models that can automate graphical user interface (GUI) layout design, a central task in modern software development such as smartphone apps.

As a designer or app developer creates an interface design, a model recommends UI elements based on the partial design that has been entered. The automation can significantly ease the effort of designers and developers for creating interfaces, because it not only reduces the input required but also brings design knowledge to the process. This is analogous to auto completion that is widely available in text editing tools, although layout completion is a much more complicated problem. The task is appealing to the deep learning field for a number of reasons. First, it is non-linear and involves 2D placement of elements, instead of the sequential next word prediction for text completion. Second, the problem is highly structural that takes structural input—a partial tree—and generates structural output—a completed tree.

For auto-completion of words, a typical choice of solution is using a language model that is trained to predict next word $w_{t+1}$ given previous words $w_{t+1}^* = \arg\max P(w_{t+1}|w_0, ..., w_t)$. As a generative model, a language model can essentially complete a sentence by predicting the rest words in an auto-regressive fashion. Intuitively, we need a layout model—a layout decoder—that predicts the rest elements and structures needed to complete a given partial tree.

Previously, models for tree structure generation have been proposed for a number of problems such as language syntax trees (Vinyals et al., 2015b) and program generation (Chen et al., 2018; Dai et al., 2018). In these problems, the model first encodes the source input via an encoder then generates tree structures in a target domain via decoding. For layout auto completion, we focus on solely involving the decoding process, although we can potentially include an encoder to bring in additional information. Additionally, our problem involves unique aspects of 2D placements.

---

[1]https://balsamiq.com, https://www.sketch.com

We design our layout decoder model based on Transformer, an model that has gained popularity on a number of tasks (Vaswani et al., 2017). The attention-based nature of Transformer allows us to easily model structures, similar to Pointer Networks (Vinyals et al., 2015a), and represent 2D spatial relationships. In particular, we examine two versions of Transformer-based tree decoder: Pointer Transformer and Recursive Transformer for this task.

Layout completion, as a structural prediction problem, lacks evaluation metrics. We design three sets of metrics to measure the quality of layout prediction based on the literature and the domain specifics of user interface interaction. We experimented with these models on a public dataset of over 50K Android user interface layouts. The experiments indicate that these models can bring values to a user interface layout design process.

## 2    RELATED WORK

Recently, there have been increasing efforts in using deep learning to enhance design practice with Generative Adversarial Networks (GANs) (Goodfellow et al., 2014), Variational Auto Encoders (Kingma and Welling, 2014), and Autoregressive models (Reed et al., 2017) as three major underlying approaches. There is a rich body of work based on Generative Adversarial Networks (GANs) (Goodfellow et al., 2014; Isola et al., 2016; Zhu et al., 2017) where a discriminator is often introduced during training for distinguishing synthesized and real designs, which in the end the model learns to generate realistic designs. In particular, LayoutGAN (Li et al., 2019) is the most related to our work in the literature, which takes a collection of randomly parameterized (e.g., positioned and sized) elements via an encoder and generates the well-arranged elements via a generator. Although we also generate 2D layouts, our problem is substantially different because it takes in a tree-structured 2D partial layout and outputs a tree-structured 2D full layout. Our model does not see all the elements as LayoutGAN does and our input and output involves a tree structure.

Our approach is related to SketchRNN (Ha and Eck, 2017), a model that can generate sketch drawings. SketchRNN takes a latent representation that is learned via Variational Inference (Kingma and Welling, 2014) and generate or complete a drawing using an autoregressive decoder (Reed et al., 2017). Similar to SketchRNN, our decoder is also an autoregressive model. The unique challenge in our case is the structure representation and generation. We can easily extend our models to take a varitional input as SketchRNN, although we focus on tree 2D decoding in this paper and leave the varitional input to future work. pix2code is another work that is related to our effort, which generates UI specifications from a screenshot pixels (Beltramelli, 2017). It uses a CNN to encode a UI screen and generate UI specification with an LSTM autoregressive decoder. In this work, a tree structure is handled as a flattened sequence such that the decoding can be handled in the same way as language sentences, which a similar treatment is conducted for syntax parse tree prediction (Vinyals et al., 2015b). Zhu et al. took one step further to propose an attention-based hierarchical decoder (Zhu et al., 2018) where two LSTMs are used: one for deciding the blocks that the program needs and the other generating tokens within a block. Our recursive Transformer is applied hierarchically as well although we use the same model repetitively.

While our target domain is user interface layouts, our problem is fundamentally related to structure prediction. Previously, much efforts have been devoted to program generation (Chen et al., 2018; Dai et al., 2018; Si et al., 2019). Jenatton et al. (Jenatton et al., 2017) proposed an approach to predict tree structures by using Bayesian optimization to combine independent Gaussian Processes with a linear model that encodes a tree-based structure. An important strategy that was explored previously is to extend inherently sequential recurrent models such as LSTM to the hierarchical situation (Tai et al., 2015), which can handle a range of tree structures. Particularly, Chen et al.'s decoder generates a binary tree by applying LSTM recursively (Chen et al., 2018), with one LSTM for generating the left child and the other for generating the right child. Dong and Lapata applied LSTM in a recursive fashion to generate logic forms from language input (Dong and Lapata, 2016). Based on the previous work, we design our recursive tree decoder based on Transformer (Vaswani et al., 2017) to handle arbitrary trees. Rather than carrying the hidden state and cell values as recurrent nets, our model can access all the nodes on the ancestry path via attention. Our positional encoding of 2D coordinates is similar to Image Transformer (Parmar et al., 2018). It is possible to employ tree positional encoding as proposed in (Shiv and Quirk, 2019) when we keep the hidden states of all the previously generated nodes. Our current design for recursive Transformer decoder is to make the ancestry hidden states

accessible for decoding child nodes. Another version of the tree decoder we experiment with in the paper is designed based on Pointer Networks (Vinyals et al., 2015a) where each node points to its parent based on the dot product similarity of their hidden states, which is an simple extension based on Transformer.

Finally, the problem we focus on is originated from the domain of human computer interaction (HCI) and graphical layout design. A rich body of works have been produced previously (Hurst et al., 2009). The idea of UI auto completion has been previously envisioned (Li and Chang, 2011). DesignScape is a full-fledged tool that provides both refinement suggestions and brainstorming suggestions during graphical layout design. Recently, Liu et al. mined a rich set of Android apps that produced over 66K UI screens with semantically labeled elements (Liu et al., 2018), including their types, bounding boxes and the hierarchical tree structures. This dataset lays the foundation for investigating the problem of structured graphical layout prediction, which constitutes the dataset for our work.

Our work is also nourished by the rich literature in the HCI domain on human performance modeling. One important aspect that is significantly underexplored for tree prediction is the accuracy metrics. We design a tree-edit distance metric based on keystroke level GOMS models (Card et al., 1980) that estimates how much effort it needs for a designer to transform the predicted layout tree into the ground-truth tree, which indicates the usefulness of a predicted layout tree. Along a few other metrics, these define a new problem that deep learning models can further tackle.

## 3 UI LAYOUT COMPLETION PROBLEM

Here, we consider the problem of completing a partial tree, which represents the graphical layout that has been entered by the designer so far (see Figure 1). A design process starts with an empty design canvas, which corresponds to a partial tree with only the root node. Each node has several properties, including its type, e.g., a button or a list, whether the node is terminal, and the rectangular bounds of the node. As the designer adds more elements to the canvas, the tree grows. Node $A$ is a parent of node $B$ if $A$ is a non-terminal node and $A$ contains $B$. The problem is extremely challenging due to the large space of possibilities.

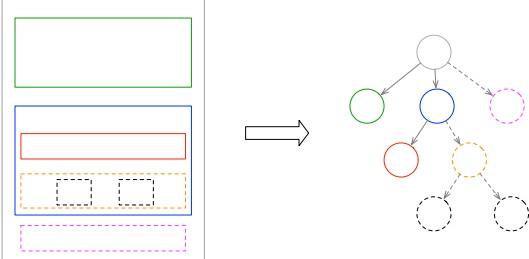

Figure 1: A schematic illustration of UI layout completion problems. The elements with solid bounding boxes are already added by the designer, and those with dashed bounding boxes are predicted to complete the layout given the solid elements—the partial layout. The tree on the right shows the UI structure with the line colors and styles corresponding the UI elements on the left. The figure is best viewed in color.

### 3.1 PROBLEM DEFINITION

A graphical layout tree, $\hat{G}$, contains a collection of $|\hat{G}|$ nodes where its $i$-th node carries both spatial and semantic properties: $(c_i, d_i, t_i, p_i, b_i), 1 \leq i \leq |\hat{G}|$ and $|\hat{G}|$ is the number of nodes in the tree. $c_i$ is a categorical value that indicates the node type, $c_i \in C$ where $C$ is the set of possible element types. $d_i$ is an integer that represents the depth of the node in the tree. $t_i$ is a binary value that represents whether the node is a terminal node. $p_i$ is the index position of the node $i$'s parent node, $1 \leq p_i \leq |\hat{G}|$. $b_i$ is a tuple that represents the bounding box of the node, including its top-left and right-bottom corners: $b_i = (x_i, y_i, \hat{x}_i, \hat{y}_i)$.

A partial tree, $\hat{P}$, contains a collection of $k$ nodes ($k < |\hat{G}|$). It is required that the root node must be in $\hat{P}$. In addition, the parent of each node in $\hat{P}$, except the root node, should also be in $\hat{P}$, i.e., $p_j \in \hat{P}$ if $j \in \hat{P}$, such that the parent index, $p_j$, is always valid and addresses an existing node in the partial tree. This requirement matches how a design tool is typically implemented where each UI element (a node) is added to either the design canvas (the root node) or a container node that is already attached to the layout tree.

Given the partial tree $\hat{P}$, the task here is to predict the full tree $\hat{G}'$ based on the partial tree: $\hat{G}' = F(\hat{P})$. We elaborate on how the function $F$ can be realized with a variety of Transformer decoders in the next section.

## 4 TRANSFORMER-BASED TREE DECODERS

Previous work for tree prediction is primarily based on LSTM, a recurrent neural network that is amenable to handle data with an arbitrary length. In this work, we design all our tree decoder models based on Transformer (Vaswani et al., 2017), an architecture that has shown advantages on a number of tasks. Particularly, the positional embedding that Transformer's attentional mechanism is based on can be easily extended to represent 2D spatial relationships. We here discuss three versions of the 2D layout tree decoder model: Vanilla, Pointer and Recursive Transformer.

### 4.1 REPRESENTING LAYOUT NODES & DECODING STEPS

Before diving into each decoder model, we first discuss the representation of each node in the layout tree, which is shared by all three decoder models. We use the general concept of embedding to represent each node (Bengio et al., 2003). More specifically, for node $i$, we first embed each property of that node as the following. For the type property, $c_i$, and the terminal property, $t_i$, we embed these categorical values as Equation 1 and 2:

$$e_i^c = \mathbb{1}(c_i)E^c \tag{1}$$

$$e_i^t = \mathbb{1}(l_i)E^t \tag{2}$$

where $e_i^c$ and $e_i^t$ are embedding for $c_i$ and $t_i$ respectively; $\mathbb{1}(\cdot)$ is a one-hot vector and $E^c \in R^{|V_c| \times |E|}$ and $E^l \in R^{|V_t| \times |E|}$ are the embedding matrix, where $|V_c|$ and $|V_t|$ are the vocabulary size of each property, and $|E|$ is the embedding dimension. For each coordinate value in $b_i$, we treat them as discrete values and represent them through a similar embedding process. See Equation 3 for embedding a coordinate $x$. A similar treatment is done previously for pixel coordinates in an image (Parmar et al., 2018).

$$e_i^x = \mathbb{1}(x_i)E^x \tag{3}$$

where $E^x \in R^{|V_x| \times |E|/4}$ and $|V_x|$ is the number of possible $x$ positions. Similarly, we embed the four coordinates in $b_i$, $x_i, y_i, \hat{x}_i, \hat{y}_i$ as $e_i^{x_i}, e_i^{y_i}, e_i^{\hat{x}_i}, e_i^{\hat{y}_i}$. We concatenate these coordinate embeddings to form the bounding box embedding (see Equation 4), which form the positional embedding for using Transformer:

$$e_i^b = [e_i^{x_i}; e_i^{y_i}; e_i^{\hat{x}_i}; e_i^{\hat{y}_i}]. \tag{4}$$

We then combine these embeddings of node properties to form the final embedding for $i$-th node using Equation 5:

$$e_i = e_i^b + e_i^c + e_i^t. \tag{5}$$

With each node represented as an embedding vector, we now discuss how the Transformer decoder model (Vaswani et al., 2017) is used to represent each decoding step. The Transformer decoder model

is a multi-layer, multihead attention-based architecture. It computes, $h_i^l$, the hidden state of step $i$ at layer $l$ by attending to the hidden states of all the steps so far in the previous layer (See Equation 9). For an $L$-layer Transformer, $0 \leq l \leq L$, and $h_i^0 = e_i$. $h_i^l \in R^{|H|}$ where $|H|$ is the dimension of the hidden state.

$$h_i^l = \text{Attention}(Q(h_i^{l-1}), K(h_{1:i}^{l-1}), V(h_{1:i}^{l-1})) \tag{6}$$

We use Attention to represent the multihead attention operation in Transformer. $Q(\cdot)$, $K(\cdot)$, and $V(\cdot)$ are feedforward nets to compute queries, keys and values for attention computation (Vaswani et al., 2017).

## 4.2 VANILLA TRANSFORMER DECODER

For the vanilla version of Transformer Tree Decoder, similar to previous work for predicting language parsing trees (Vinyals et al., 2015b), we linearize a tree, based on the preorder depth-first traversal, as a sequence of tokens. The process adds the opening "(" and closing ")" tokens for the children of each parent except the case when the parent has a single child, as illustrated in Figure 2 in (Vinyals et al., 2015b). Note that this representation requires the partial tree to be a prefix of the depth-first traversal. This data representation transforms tree prediction to a sequence prediction problem.

We can then define the distribution over a sequence of $m$ decoded tree nodes, $n_j, k+1 \leq j \leq k+m$, given a sequence of $k$ nodes from a given partial tree $\hat{P}$: $n_j, 1 \leq j \leq k$, as the following (see Equation 7).

$$
\begin{aligned}
P(n_{k+1}, n_{k+2}, ..., n_{k+m} | n_1, n_2, ..., n_k) &= \prod_{i=k+1}^{k+m} P(n_i | n_1, n_2, ..., n_k, n_{k+1}, ..., n_{i-1}) \\
&= \prod_{i=k+1}^{k+m} P(c_i, t_i, x_i, y_i, \hat{x}_i, \hat{y}_i | n_1, n_2, ..., n_k, n_{k+1}, ..., n_{i-1}) \\
&= \prod_{i=k+1}^{k+m} \prod_{z \in \{c_i, t_i, x_i, y_i, \hat{x}_i, \hat{y}_i\}} \text{softmax}(W_z h_i^L)
\end{aligned}
\tag{7}
$$

where $n_i = (c_i, t_i, x_i, y_i, \hat{x}_i, \hat{y}_i)$. $W_z \in R^{|V_z| \times |H|}$ is the output embedding weights for each property of a node where $|V_z|$ is the vocabulary size for the property and $|H|$ is the embedding dimension. The model does not directly predict $p_i$. Instead, $p_i$ is acquired by reconstructing the tree from brackets. Note that the opening and closing bracket are considered two special nodes that need to be embedded and predicted as well. For these two nodes, only $c_i$ matters while other node properties are irrelevant.

## 4.3 POINTER TRANSFORMER DECODER

Instead of introducing the beginning and closing tokens to represent tree hierarchies and being limited to only the depth-first traversal order, Pointer Networks (Vinyals et al., 2015a) provide a natural way to represent child-parent relationship. We define $n_i' = (c_i, t_i, x_i, y_i, \hat{x}_i, \hat{y}_i, p_i)$ that includes the index position of the node's parent, $p_i$. We then extend Equation 7 to predict $p_i$ as the following (see Equation 8).

$$
\begin{aligned}
P(n_{k+1}', ..., n_{k+m}' | n_1, ..., n_k) &= \prod_{i=k+1}^{k+m} P(c_i, t_i, x_i, y_i, \hat{x}_i, \hat{y}_i, p_i | n_1, ..., n_k, n_{k+1}, ..., n_{i-1}) \\
&= \prod_{i=k+1}^{k+m} [\text{softmax}(H_{<i}^L h_i^L) \prod_{z \in \{c_i, t_i, x_i, y_i, \hat{x}_i, \hat{y}_i\}} \text{softmax}(W_z h_i^L)]
\end{aligned}
\tag{8}
$$

where $H_{<i}^L \in R^{(i-1)\times|H|}$ is the embedding weights with $j$th row be $h_j^L, 1 \leq j \leq i-1$. softmax$(H_{<i}^L h_i^L)$ computes a softmax over the dot product alignments between the hidden state of the $i$th node and that of each previous node.

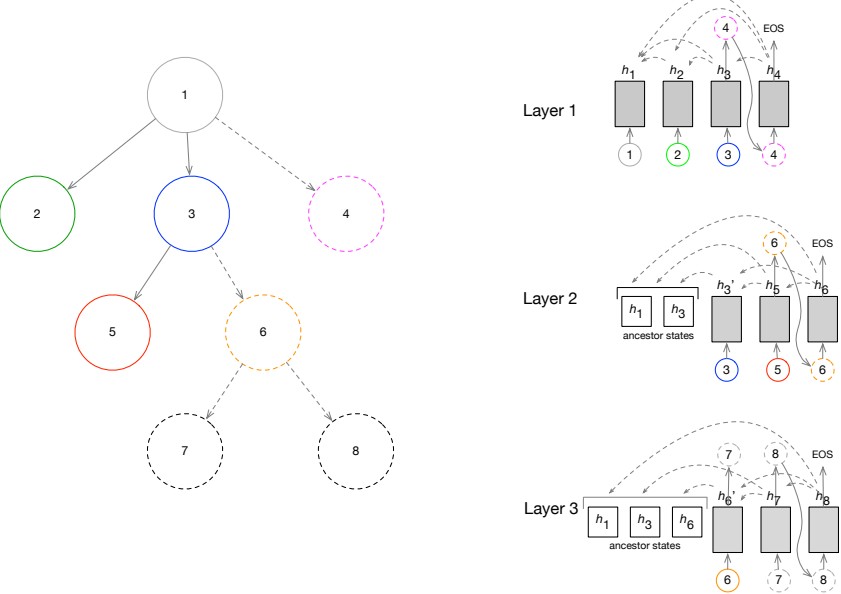

Figure 2: A schematic illustration of Recursive Transformer for decoding the tree on the left. The same Transformer model is applied to decoding the children for each parent, top-down from the root. The hidden states of ancestry nodes are used for computing attention in downstream layers. The gray boxes represent Transformer layers and the dashed lines denote attention dependencies. The dashed nodes are predicted.

## 4.4 RECURSIVE TRANSFORMER DECODER

The third version of decoder models that we investigate here applies a Transformer decoder model in a recursive manner by using the same model to decode the children of each parent node (see Figure 2). To do so, we first compute the decoder self attention in a similar way to Equation 9, except that we only use the parent and sibling nodes for computing attention keys and values. $\tilde{h}_s^l$ represents the hidden state after taking the $s$th sibling node.

$$\tilde{h}_s^l = \text{Attention}(Q(h_s^{l-1}), K(h_{1:s}^{l-1}), V(h_{1:s}^{l-1})) \tag{9}$$

To leverage the information beyond the parent node and siblings, we involve the ancestry nodes in the attention computation by attending to their hidden states.

$$h_s^l = \text{Attention}(Q(\tilde{h}_s^l), K(H_\mathcal{A}^L), V(H_\mathcal{A}^L)) \tag{10}$$

where $\mathcal{A}$ donates the set of ancestry nodes, and $H_\mathcal{A}^L$ are the set of hidden states of the ancestry nodes, which have been previously computed as the decoding is performed in a top-down manner starting from the root node. The first node fed into the decoder is the parent node. The hidden states $h_s^L$ capture the information beyond the ancestry nodes as these nodes were able to access other nodes in the tree during attention computation. We can then define the distribution over a sequence of sibling nodes given the partial tree $\hat{P}$ in a similar fashion to Equation 7.

The process not only decodes siblings but also produces hidden states that will be accessed in the downstream layers of the tree as ancestry states. The decoding process starts from decoding the the children of the root node, because the root node that corresponds the blank design canvas is always given. For each non-terminal nodes, $t_i = 0$, we apply the same transformer model to decode its

children. The size of ancestry state grows linearly with respect to the depth of the tree (see the schema illustration in Figure 2). Note that in this model, we do not flatten the tree representation and we do not compute parent pointer either. This design allows us to easily batch the training and decoding for multiple trees at the same time, where multiple trees can be treated as a forest.

## 5 EVALUATION

In this section, we evaluate the tree decoders we propose for UI layout completion. We start by describing the dataset we use and elaborate on data processing we conducted. We then define the metrics for this new problem. Lastly, we report on the results of our experiments.

### 5.1 DATASETS

We use a public dataset that includes over 60K UI layouts where each element in a layout is labeled with a set of properties including its type and bounding boxes (Liu et al., 2018). These screens and hierarchies capture 25 UI component categories, such as text, buttons lists and icons. For each layout, a corresponding UI hierarchy represents the tree structure of the UI.

We preprocess the data to filter out ill-formed layouts. When a node is out of screen or outside its parent, we remove the entire layout out of the dataset. We also filter out long-tail examples when a layout has more than 100 nodes in the tree or a parent has more than 30 children. These long-tail examples are a small portion of the entire dataset. The resulted dataset has more than 55K examples. There are 25 type categories, the average number of nodes per layout is 16 (max=49 and min=2), and the average depth of each layout tree is 3 (max=5, min=2). We also scale down the coordinate values to the range of 72x128 for the horizontal and vertical dimensions. At this scale, a layout is still readable for human eyes.

### 5.2 METRICS

As far as we know, there are no established metrics for the problem of layout design completion. To evaluate the quality of each decoder model, we propose three metrics, which address different user scenarios.

**Layout Tree Edit Distance**  The purpose of this metric is to estimate how much effort is required for a designer to transform a predicted layout into the target one. We implement this metric based on optimal tree edit distance calculation (Zhang and Shasha, 1989), and ground each tree manipulation cost into time effort based on Keystroke-level GOMS model (Card et al., 1980; Kieras, 2019). There are three edit operators in the edit distance calculation: Insertion, Deletion and Change. For Keystroke-level GOMS analysis, we assume a typical layout editing environment with all the UI elements presented in a palette. For Insertion, the effort involves the designer selecting a target element from the palette, dragging and dropping it onto the design canvas and then resizing it to a target dimension. For Change, if the type is incorrect, the designer needs to right click on the element and then select a target type. If the location or size is different, the designer needs to correct them by dragging and dropping as well. Lastly, for Deletion, since the designer can ignore the predictions, the cost is set to minimal. Note that these cost analyses should not be treated as an absolute measure of effort. Rather, this metric should be considered as a relative measure to compare models.

**Parent-Child Pair Retrieval Accuracy**  The other measure we propose is to treat the task as an information retrieval problem (IR) in which we measure how well the model can predict parent-child pairs against the set of pairs in the ground-truth tree. This metric is less sensitive to the overall tree structure and is only concerned with local layout structures. A parent-child node pair, $(p, s)$, is defined as a concatenation of their type and spatial properties, $(c_p, x_p, y_p, \hat{x}_p, \hat{y}_p, c_s, x_s, y_s, \hat{x}_s, \hat{y}_s)$. A pair is successfully retrieved only if all the values in the predicted tuple match a ground-truth tuple. With this formation, we can compute scores such as precision, recall and their combination F1 scores.

**Next-Element Prediction Accuracy**  Because it is challenging to predict the entire tree, we look into the metrics that can capture how well a model predicts next element the designer needs. This is analogous to next word prediction in language models. For this metric, given a prefix of the

preorder or breadth-first traversal sequence of the ground-truth layout tree, we test if the next element, immediately following the prefix, is predicted by the model.

## 5.3 Experiments

**Model Configuration & Training**    We implemented these models in TensorFlow based on the Transformer implementation in Tensor2Tensor[2]. We split the dataset for training (80%), validation (10%) and test (10%). Based on the training and the validation datasets, we determined an optimal model architecture and hyperparameters, including finding the hidden size in the range of [128, 256, 512] and the number of layers [2, 4, 6] as well as tuning on the learning rates and dropout ratios. We trained each of the models on a single machine with 8 Tesla P100 or V100 GPU cores, using a batch size of 128—the number layout trees. The training strategy is similar to the one introduced by Transformer (Vaswani et al., 2017). We trained these models until they converge.

**Partial Tree Setups**    Our models do not require a design process to be carried out in a certain order. However, to examine how these models would aid a design, we need to assume a variety of design flows that a designer might follow in a realistic design process, which would lead to different layout trees.

One factor is the order that the designer wants to progress a design structurally: the depth-first versus the breadth-first strategies. For the depth-first strategy, the designer starts by adding the container elements (parent nodes) and then finishes the child elements in the container before moving onto the next group of elements. For the breadth-first strategy, the designer will layout top-level elements before adding more detailed designs to each container. In reality, a designer might alternate between the two, which we leave for future evaluation. All these strategies are assumed for examining the models although the models can be used for completing a design of an arbitrary order.

For each of these flows, we vary the size of the partial layout given to the model, including 10%, 50% and 80% of the full tree. These correspond to the portion of a layout design that has been already entered by the designer. Given a partial tree, there can be more than one way to complete the layout. Given a 10%, 50% and 80% BFS partial layout, the mean number of completions of the layout is 2.97, 1.23 and 1.17 respectively. Given a 10%, 50% and 80% DFS partial layout, the mean number of completions is 3.63, 1.24, and 1.17 respectively.

**Results**    As we can see, Recursive Transformer Decoder outperforms the Pointer Decoder for most cases when the design flow follows the breadth-first traversal (see Table 3), even though Pointer Decoder has the direct access to all the partial tree and previously decoded nodes and Recusive Decoder does not. Vanilla Decoder is not tested for the BFS case because it is only structured for DFS as discussed earlier. For the depth-first traversal case (see Table 2), both Pointer and Recursive outputform the Vanilla version with a large margin. However, the trend between Pointer and Recursive decoders is less obvious than the BFS case. This is understandable because the Recursive Transformer decodes in a top-down manner, which cannot leverage the given nodes in the partial tree in deeper layers. The depth-first traversal tree put Recursive decoder in a disadvantageous position. In contrast, Pointer decoder has direct access to all the nodes in the given partial tree. Nevertheless, Recursive Transformer still consistently outperforms Pointer Transformer on Next Item prediction with a large margin.

Table 1: The prediction accuracy for the breadth-first layout design flow. The F1 score (%) for retrieving parent-child pairs, next item accuracy (%), and Edit distances are calculated given 10%, 50% and 80% of the ground-truth tree as the partial tree. For F1 and Next Item Accuracy, the larger is the better, while for Edit Distance, the smaller the better.

| Models | BFS 10% | | | BFS 50% | | | BFS 80% | | |
|---|---|---|---|---|---|---|---|---|---|
| | F1 | Next | Edit | F1 | Next | Edit | F1 | Next | Edit |
| Pointer | 11.19 | 12.25 | 87.40 | 61.31 | 16.76 | **35.78** | **87.24** | 27.56 | **13.55** |
| Recursive | **15.23** | **18.4** | **71.26** | **64.07** | **26.84** | 46.47 | 86.41 | **36.58** | 27.22 |

[2]https://github.com/tensorflow/tensor2tensor

Table 2: The prediction accuracy for the depth-first layout design flow. The F1 score (%), next item accuracy (%), and Edit distances are calculated given 10%, 50% and 80% of the ground-truth tree.

| Models | DFS 10% | | | DFS 50% | | | DFS 80% | | |
|---|---|---|---|---|---|---|---|---|---|
| | F1 | Next | Edit | F1 | Next | Edit | F1 | Next | Edit |
| Vanilla | 5.35 | 0.74 | 234.17 | 40.4 | 5.35 | 138.88 | 68.82 | 17.03 | 84.32 |
| Pointer | 13.12 | 20.21 | 76.44 | **65.49** | 12.03 | **21.41** | **88.96** | 22.38 | **9.13** |
| Recursive | **15.05** | **29.02** | **62.49** | 60.79 | **22.93** | 52.26 | 84.65 | **33.16** | 34.24 |

Table 3: The prediction accuracy for the breadth-first layout design flow with a random spatial ordering. The F1 score (%) for retrieving parent-child pairs, next item accuracy (%), and Edit distances are calculated given 10%, 50% and 80% of the ground-truth tree as the partial tree. For F1 and Next Item Accuracy, the larger is the better, while for Edit Distance, the smaller the better.

| Models | BFS 10% | | | BFS 50% | | | BFS 80% | | |
|---|---|---|---|---|---|---|---|---|---|
| | F1 | Next | Edit | F1 | Next | Edit | F1 | Next | Edit |
| Pointer | 11.19 | 12.25 | 87.40 | 61.31 | 16.76 | **35.78** | **87.24** | 27.56 | **13.55** |
| Recursive | **15.23** | **18.4** | **71.26** | **70.0** | **17.0** | 22.46 | 91.8 | **30.47** | 11.79 |

## 6 DISCUSSIONS

In this paper, we introduce a new problem of auto completion for UI layout design. We formulate the problem as partial tree completion, and investigate a range of variations of layout decoders based on Transformer. The two models we proposed, Pointer and Recursive Transformer, gave reasonable predictions (see Figure 3). We also define task setups and evaluation metrics for examining the quality of prediction and report the results based on a public dataset.

While both Pointer and Recursive Transformer clearly outperformed the baseline model, we found that layout completion remains a challenging task. Our model is often able to predict layout structures and semantic properties well, but less accurate on bounding boxes. There are many UIs with the same layout structure but different spatial details, while our current eval is using hard metrics. When relaxing the metrics by only considering structures and semantic properties, all the models got much better accuracy, e.g., Next item for 80% BFS partial (Recursive: 95.3%, Pointer: 92.9%) and for 80% DFS partial (Recursive: 93.4%, Pointer: 88.4%, Vanilla: 76.6%). Recursive still shows significant advantages over other models. We experimented with continuous coordinate output using squared errors for loss. The results showed that the model with continuous coordinates performs substantially worse than treating coordinates as one-hot categorical values, which deserves further investigation.

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

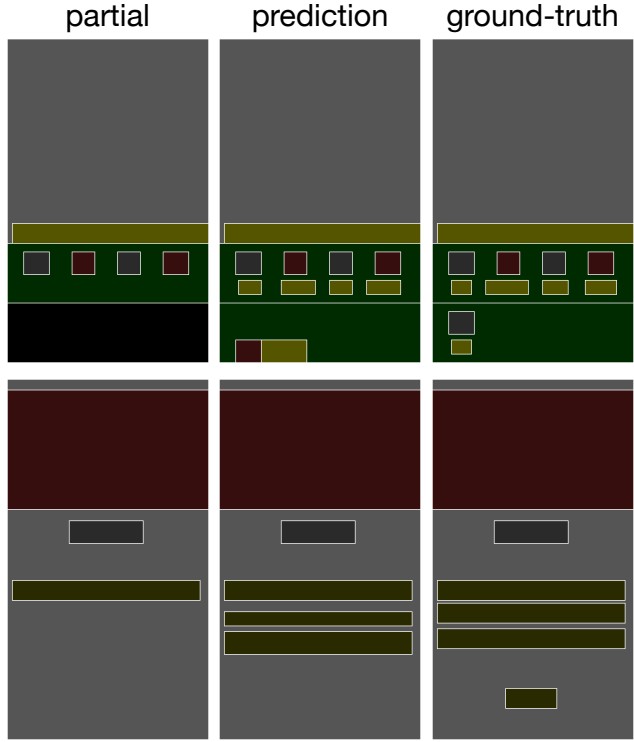

Figure 3: Examples of predicted layouts given a partial layout, with respect to the corresponding groundtruth complete layout.

H. Dai, Y. Tian, B. Dai, S. Skiena, and L. Song. Syntax-directed variational autoencoder for structured data. *CoRR*, abs/1802.08786, 2018. URL http://arxiv.org/abs/1802.08786.

L. Dong and M. Lapata. Language to logical form with neural attention. *CoRR*, abs/1601.01280, 2016. URL http://arxiv.org/abs/1601.01280.

I. Goodfellow, J. Pouget-Abadie, M. Mirza, B. Xu, D. Warde-Farley, S. Ozair, A. Courville, and Y. Bengio. Generative adversarial nets. In Z. Ghahramani, M. Welling, C. Cortes, N. D. Lawrence, and K. Q. Weinberger, editors, *Advances in Neural Information Processing Systems 27*, pages 2672–2680. Curran Associates, Inc., 2014. URL http://papers.nips.cc/paper/5423-generative-adversarial-nets.pdf.

D. Ha and D. Eck. A neural representation of sketch drawings. *CoRR*, abs/1704.03477, 2017. URL http://arxiv.org/abs/1704.03477.

N. Hurst, W. Li, and K. Marriott. Review of automatic document formatting. In *Proceedings of the 9th ACM Symposium on Document Engineering*, DocEng '09, pages 99–108, New York, NY, USA, 2009. ACM. ISBN 978-1-60558-575-8. doi: 10.1145/1600193.1600217. URL http://doi.acm.org/10.1145/1600193.1600217.

P. Isola, J. Zhu, T. Zhou, and A. A. Efros. Image-to-image translation with conditional adversarial networks. *CoRR*, abs/1611.07004, 2016. URL http://arxiv.org/abs/1611.07004.

R. Jenatton, C. Archambeau, J. González, and M. Seeger. Bayesian optimization with tree-structured dependencies. In D. Precup and Y. W. Teh, editors, *Proceedings of the 34th International Conference on Machine Learning*, volume 70 of *Proceedings of Machine Learning Research*, pages 1655–1664, International Convention Centre, Sydney, Australia, 06–11 Aug 2017. PMLR. URL http://proceedings.mlr.press/v70/jenatton17a.html.

D. Kieras. Goms models for task analysis. 05 2019.

D. P. Kingma and M. Welling. Auto-encoding variational bayes. In *2nd International Conference on Learning Representations, ICLR 2014, Banff, AB, Canada, April 14-16, 2014, Conference Track Proceedings*, 2014. URL `http://arxiv.org/abs/1312.6114`.

J. Li, J. Yang, A. Hertzmann, J. Zhang, and T. Xu. Layoutgan: Generating graphic layouts with wireframe discriminators. *CoRR*, abs/1901.06767, 2019. URL `http://arxiv.org/abs/1901.06767`.

Y. Li and T.-H. Chang. Auto-completion for user interface design. US Patent, 2011. URL `https://patents.google.com/patent/US20150169140`.

T. F. Liu, M. Craft, J. Situ, E. Yumer, R. Mech, and R. Kumar. Learning design semantics for mobile apps. In *Proceedings of the 31st Annual ACM Symposium on User Interface Software and Technology*, UIST '18, pages 569–579, New York, NY, USA, 2018. ACM. ISBN 978-1-4503-5948-1. doi: 10.1145/3242587.3242650. URL `http://doi.acm.org/10.1145/3242587.3242650`.

N. Parmar, A. Vaswani, J. Uszkoreit, L. Kaiser, N. Shazeer, A. Ku, and D. Tran. Image transformer. In J. Dy and A. Krause, editors, *Proceedings of the 35th International Conference on Machine Learning*, volume 80 of *Proceedings of Machine Learning Research*, pages 4055–4064, Stockholmsmässan, Stockholm Sweden, 10–15 Jul 2018. PMLR. URL `http://proceedings.mlr.press/v80/parmar18a.html`.

S. E. Reed, A. van den Oord, N. Kalchbrenner, S. G. Colmenarejo, Z. Wang, D. Belov, and N. de Freitas. Parallel multiscale autoregressive density estimation. *CoRR*, abs/1703.03664, 2017. URL `http://arxiv.org/abs/1703.03664`.

V. L. Shiv and C. Quirk. Novel positional encodings to enable tree-structured transformers, 2019. URL `https://openreview.net/forum?id=SJerEhR5Km`.

X. Si, Y. Yang, H. Dai, M. Naik, and L. Song. Learning a meta-solver for syntax-guided program synthesis. In *International Conference on Learning Representations*, 2019. URL `https://openreview.net/forum?id=Syl8Sn0cK7`.

K. S. Tai, R. Socher, and C. D. Manning. Improved semantic representations from tree-structured long short-term memory networks. *CoRR*, abs/1503.00075, 2015. URL `http://arxiv.org/abs/1503.00075`.

A. Vaswani, N. Shazeer, N. Parmar, J. Uszkoreit, L. Jones, A. N. Gomez, L. Kaiser, and I. Polosukhin. Attention is all you need. *CoRR*, abs/1706.03762, 2017. URL `http://arxiv.org/abs/1706.03762`.

O. Vinyals, M. Fortunato, and N. Jaitly. Pointer networks. In C. Cortes, N. D. Lawrence, D. D. Lee, M. Sugiyama, and R. Garnett, editors, *Advances in Neural Information Processing Systems 28*, pages 2692–2700. Curran Associates, Inc., 2015a. URL `http://papers.nips.cc/paper/5866-pointer-networks.pdf`.

O. Vinyals, L. Kaiser, T. Koo, S. Petrov, I. Sutskever, and G. Hinton. Grammar as a foreign language. In *Proceedings of the 28th International Conference on Neural Information Processing Systems - Volume 2*, NIPS'15, pages 2773–2781, Cambridge, MA, USA, 2015b. MIT Press. URL `http://dl.acm.org/citation.cfm?id=2969442.2969550`.

K. Zhang and D. Shasha. Simple fast algorithms for the editing distance between trees and related problems. *SIAM J. COMPUT*, 18(6), 1989.

X. Zheng, X. Qiao, Y. Cao, and R. W. H. Lau. Content-aware generative modeling of graphic design layouts. *ACM Trans. Graph.*, 38(4):133:1–133:15, July 2019. ISSN 0730-0301. doi: 10.1145/3306346.3322971. URL `http://doi.acm.org/10.1145/3306346.3322971`.

J. Zhu, T. Park, P. Isola, and A. A. Efros. Unpaired image-to-image translation using cycle-consistent adversarial networks. *CoRR*, abs/1703.10593, 2017. URL `http://arxiv.org/abs/1703.10593`.

Z. Zhu, Z. Xue, and Z. Yuan. Automatic graphics program generation using attention-based hierarchical decoder. *CoRR*, abs/1810.11536, 2018.

