# OpenReview forum: "Auto Completion of User Interface Layout Design Using Transformer-Based Tree Decoders"
_ICLR.cc/2020/Conference — Reject_

### Official Review · AnonReviewer3 · 2019-10-21
**Official Blind Review #3**

**Rating:** 3

**Review:**

The paper presents an auto completion for UI layout design. The authors formulate the problem as partial tree completion, and investigate a range of variations of layout decoders based on Transformer.

The paper proposes two models: Pointer and Recursive Transformer. The paper designs three sets of metrics to measure the quality of layout prediction based on the literature and the domain specifics of user interface interaction.

The writing quality is readable. The presentation is nice. The task of auto completion for UI layout design is relatively new.

The paper misses the key baseline in Bayesian optimisation using tree structure [1] which can perform the prediction under the tree-structure dependencies.

[1] Jenatton, Rodolphe, et al. "Bayesian optimization with tree-structured dependencies." Proceedings of the 34th International Conference on Machine Learning-Volume 70. JMLR. org, 2017.

NB: the reviewer has low confidence in evaluating this paper.



**Experience Assessment:**

I have read many papers in this area.

**Review Assessment: Checking Correctness Of Derivations And Theory:**

I did not assess the derivations or theory.

**Review Assessment: Checking Correctness Of Experiments:**

I assessed the sensibility of the experiments.

**Review Assessment: Thoroughness In Paper Reading:**

I made a quick assessment of this paper.

---

> ### Author Response · Authors · 2019-11-15
> **Comparison with Jenatton, Rodolphe, et al.'s work**
>
> Thanks for your comments. We missed this previous work. Jenatton et al. proposed an approach to predict tree structures by using Bayesian optimization to combine independent Gaussian Processes with a linear model that encodes a tree-based structure. We have cited and discussed the work in the revision. The focus of our work is to propose a new tree prediction problem (layout completion) and introduce Transformer-based approaches for addressing the problem. It would be future work to investigate other tree-based models for this problem.

---

### Official Review · AnonReviewer1 · 2019-10-24
**Official Blind Review #1**

**Rating:** 1

**Review:**

This paper proposes an autocompletion model for UI layout based on adaptations of Transformers for tree structures and evaluates the models based on a few metrics on a public UI dataset.

I like the area of research the authors are looking into and I think it's an important application. However, the paper doesn't answer key questions about both the application and the models:

1) There is no clear rationale on why we need a new model based on Transformers for this task. What was wrong with LSTMs/GRUs as they've been used extensively for recursive problems including operations on trees? Similarly, I'd have expected baselines that included those models in the evaluation section showing the differences in performance between the newly proposed Transformer model for trees and previously used methods.

2) The evaluation metrics used while borrowed from the language or IR fields doesn't seem to translate to UI design. UI layout is about visual and functional representation of an application so if one is seeking to evaluate different models, they need to relate to those.

**Experience Assessment:**

I have read many papers in this area.

**Review Assessment: Checking Correctness Of Derivations And Theory:**

I carefully checked the derivations and theory.

**Review Assessment: Checking Correctness Of Experiments:**

I carefully checked the experiments.

**Review Assessment: Thoroughness In Paper Reading:**

I read the paper at least twice and used my best judgement in assessing the paper.

---

> ### Author Response · Authors · 2019-11-15
> **LSTM and Eval Metrics**
>
> Thank you for your comments.
>
> - LSTM
> LSTMs are indeed a strong model for tree prediction on previous tasks. To allow the model to access ancestry nodes during decoding, one way is to concatenate the parent node latent representation with the input of each step for decoding children, and then feed the concatenated vector to LSTM (e.g., Dong & Lapata ACL 2016). However, since the ancestry has a variable-number of nodes (as decoding proceeds), to directly access these nodes during decoding, attentional mechanisms would be an efficient way, which is one of our motivations to use Transformer models that are attention-based. Of course, LSTM equipped with Attention would achieve the same benefit. In addition, positional encoding in Transformer also allows us to easily model spatial locations of UI elements. Our early experiments with LSTM did not yield good results on this spatial layout problem. That said, we agree it is worth investigating the performance of LSTM on this problem further. Since this is the first paper on this topic, we chose to focus on introducing the problem and providing Transformer-based approaches as a baseline for future work.
>
> - Eval metrics
> We agree the IR-based metrics have limitations. This is why we provided multiple eval metrics including edit distances and next-N accuracy. The Edit Distance metric was designed by taking into account human factors in interaction tasks based on the key-stroke level GOMS models. We can clarify this further in the revision.

---

### Official Review · AnonReviewer2 · 2019-10-25
**Official Blind Review #2**

**Rating:** 3

**Review:**

Summary: This paper introduces the task of using deep learning for auto-completion in UI design. The basic idea is that given a partially completed tree (representing the design state of the UI), the goal is to predict or "autocomplete" the final tree. The authors propose a transformer-based solution to the task, considering three variants: a vanilla approach where the tree is flattened to a sequence, a pointer-network style approach, and a recursive transformer. Preliminary experiments indicate that the recursive model performs best and that the task is reasonable difficulty.

Assessment: Overall, this is a borderline paper, as the task is interesting and novel, but the presentation is lacking in technical detail and there is a lack of novelty on the modeling side.

In particular, the authors spend a bulk of the paper describing the three different baselines they implement. However, despite the fact that most of the paper is dedicated to the explanation of these baselines. There is not sufficient detail to reproduce the models based on the paper alone. Indeed, without referencing the original Pointer Network and (and especially the) Transformer papers, it would not be possible to understand this paper at all. Further technical background and detail would drastically improve the paper. Moreover, it seems strange that significant space was used to give equations describing simple embedding lookups (i.e., matrix multiplications with one-hot vectors), but the basic technical foundations of Transformers were not adequately explained.  In addition, only the transformer baselines were considered, and it would seem natural to consider LSTM-based baselines, or some other related techniques.  In general, the space that was used to explain the Transformer baselines---which are essentially straightforward ways to adapt transformers to this task---could have been used to give more detail on the dataset. For example, one question is how often a single partial tree has multiple possible completions in the data.

A major issue---mainly due to the lack of technical details and the lack of promise to provide code/data (unless I missed this)---is that the paper does not appear to be reproducible. Given the intent to have this be a new benchmark, ensuring reproducibility seems critical.

Reasons to accept:
- Interesting new application of GNNs

Reasons to reject:
- Incremental modeling contribution
- Lack of sufficient technical detail on models and dataset
- Does not appear to be reproducible



**Experience Assessment:**

I have read many papers in this area.

**Review Assessment: Checking Correctness Of Derivations And Theory:**

I assessed the sensibility of the derivations and theory.

**Review Assessment: Checking Correctness Of Experiments:**

I assessed the sensibility of the experiments.

**Review Assessment: Thoroughness In Paper Reading:**

I read the paper at least twice and used my best judgement in assessing the paper.

---

> ### Author Response · Authors · 2019-11-15
> **Re: Contribution, Reproducible and Technical Details**
>
> Thank you for your comments. We have revised the paper to address the issues you brought up.
>
> - Contribution
> We developed our approach based on Transformer models. We agree with the reviewer that the model novelty is relatively incremental. However, the focus of the paper is to contribute a new prediction problem and adapts and applies the Transformer model for this problem to establish a benchmark for future exploration, which we believe has values.
>
> - Benchmark & Reproducible
> The data that our experiments used is an open dataset:
> https://storage.cloud.google.com/crowdstf-rico-uiuc-4540/rico_dataset_v0.1/semantic_annotations.zip
> We will release our data preprocessing, and model code, including all the eval metrics to ensure the work is reproducible.
>
> - Technical details
> Thanks for pointing out the issues with our presentations. We agree much detail on embeddings can be condensed or moved to Appendix. We included embedding details in the paper because a reviewer from the venue we previously submitted to requested these details to be in the paper. We revised the notations in the paper to make formulation clearer. In addition, we added more details about the data as you suggested. Given a partial tree, there can be more than one way to complete the layout. Given a 10%, 50% and 80% BFS partial layout, the mean number of completions of the layout is 2.97, 1.23 and 1.17 respectively. Given a 10%, 50% and 80% DFS partial layout, the mean number of completions is 3.63, 1.24, and 1.17 respectively.

---

### Decision · Program_Chairs · 2019-12-19

**Decision:**

Reject

**Comment:**

The paper introduces an interesting application of GNNs, but the reviewers find that the contribution is too limited and the motivation is too weak.